# CoFrNets: Interpretable Neural Architecture Inspired by Continued Fractions

**Isha Puri**[*]
Harvard University
ishapuri@college.harvard.edu

**Amit Dhurandhar**
IBM Research
adhuran@us.ibm.com

**Tejaswini Pedapati**
IBM Research
tejaswinip@us.ibm.com

**Karthikeyan Shanmugam**
IBM Research
karthikeyan.shanmugam2@ibm.com

**Dennis Wei**
IBM Research
dwei@us.ibm.com

**Kush R. Varshney**
IBM Research
krvarshn@us.ibm.com

## Abstract

In recent years there has been a considerable amount of research on local post hoc explanations for neural networks. However, work on building interpretable neural architectures has been relatively sparse. In this paper, we present a novel neural architecture, CoFrNet, inspired by the form of continued fractions which are known to have many attractive properties in number theory, such as fast convergence of approximations to real numbers. We show that CoFrNets can be efficiently trained as well as interpreted leveraging their particular functional form. Moreover, we prove that such architectures are universal approximators based on a proof strategy that is different than the typical strategy used to prove universal approximation results for neural networks based on infinite width (or depth), which is likely to be of independent interest. We experiment on nonlinear synthetic functions and are able to accurately model as well as estimate feature attributions and even higher order terms in some cases, which is a testament to the representational power as well as interpretability of such architectures. To further showcase the power of CoFrNets, we experiment on seven real datasets spanning tabular, text and image modalities, and show that they are either comparable or significantly better than other interpretable models and multilayer perceptrons, sometimes approaching the accuracies of state-of-the-art models.

## 1 Introduction

"It is simple. The minute I heard the problem, I knew that the answer was a continued fraction. Which continued fraction, I asked myself. Then the answer came to my mind."

This was the response of the mathematics genius Ramanujan to Mahalanobis, who was astounded how he was able to solve the difficult Strand puzzle [30] almost instantaneously. Besides showcasing the genius of Ramanujan, the puzzle also showcases the power of continued fractions. Continued fractions (CFs), typically represented as a sequence that looks like a ladder: $a_0 + \cfrac{b_1}{a_1 + \cfrac{b_2}{a_2 + \cdots}}$, can represent any real number and any analytic function, including trignometric functions, polynomials, the exponential function, power functions, and special functions like the gamma, hypergeometric, and Bessel functions [8]. To represent arbitrary real numbers, it is sufficient for the $a_k$s to be non-negative integers and for the $b_k = 1$. Analytic functions, which can be represented as a power series, can also

---
[*]Work done as part of internship at IBM Research.

35th Conference on Neural Information Processing Systems (NeurIPS 2021).

be represented in this form. Moreover, CFs are the best rational approximations to a number/function in a certain sense [31]. Rational approximations are obtained by curtailing the fraction just before the "+" sign. For instance, in the example CF above $a_0$, $\frac{a_0 a_1 + b_1}{a_1}$, and $\frac{a_0 a_1 a_2 + a_0 b_2 + a_2 b_1}{a_1 a_2 + b_2}$ are three different rational approximations. Additional properties of CFs are discussed in Section 3.

Given the desirable properties of CFs and noticing their ladder-like structure, we propose a neural architecture inspired by CFs illustrated in Figure 1. In place of the $a_k$s, linear functions of the input $x \in \mathbb{R}^p$ are computed by taking the inner product of $x$ with weight vector $w_k \in \mathbb{R}^p$ in each layer $k$ (or step of the ladder).[2] The reciprocal of the function thus far is applied as a nonlinearity in each layer. We refer to this **Co**ntinued **Fr**action-inspired neural network as CoFrNet (pronounced 'coffer net'). (Like coffers in building architecture [49], the proposed neural architecture is made up of repeating structures.) Although more complicated functions than linear could be used in each layer, we show in Section 5 that linear functions are sufficient for universal approximation with a finite number of such ladders. The proof follows a different strategy than typical results on universal approximation of neural networks that rely on the results of Cybenko [9], Hornik [21], and Zhou [27], and may be of independent interest.

The proposed architecture is "simple": there are only $p$ weights to be learned in each layer as opposed to a quadratic number for standard architectures such as multilayer perceptron (MLP) or those with densely connected layers.[3] A key differentiation from other neural architectures is that the input is passed into every layer [15, 19, 16, 47].[4] Moreover, the nonlinearity $\frac{1}{z}$ is different from more commonly used nonlinearities such as sigmoid, ReLU, and polynomials. The CF representation is much more compact than directly representing a polynomial of the same degree, which requires exponentially many coefficients. We later show how the CF representation for analytic functions permits the architecture to be made human-interpretable.

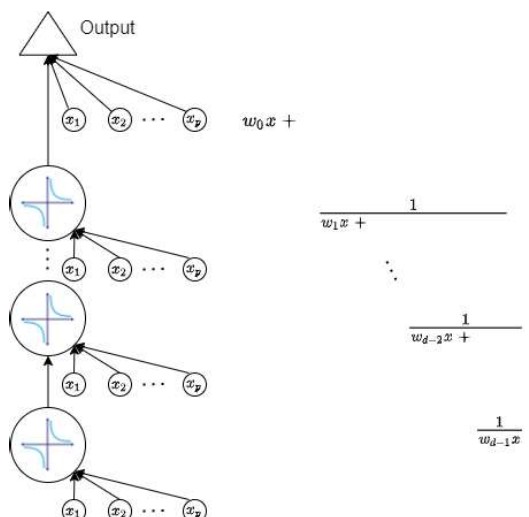

Figure 1: Single ladder (depth $d$) CoFrNet architecture on the left. On the right we see the corresponding function computed at each stage.

Simply being able to represent a rich class of functions does not imply effective learnability. (After all, unlike Ramanujan, the answer will not simply come to the machine's mind.) However, we empirically demonstrate that learning this function class is indeed possible. We propose variants of the base architecture catered toward ease of interpretation and efficiency of training, while still minimizing generalization error. We apply CoFrNets to tabular, text, and image data, and show they are either competitive or significantly better than other interpretable models and MLPs. In addition, we are not only able to model synthetic data generated from complicated nonlinear functions accurately, but also obtain feature attributions and recover the functional form reasonably well by leveraging properties of the architecture.

In summary, the main contribution of this paper is a new architecture covering an interesting and rich function class. By taking advantage of properties studied from the very beginnings of formal mathematics, we have stumbled upon a simple, yet powerful new idea in neural architecture design with much promise for accuracy, interpretability, and even efficiency. In this initial paper, we believe we have only scratched the surface of the CoFrNet architecture. Different training strategies and architecture variants as discussed briefly in Sections 4 and 7, among other enhancements, may lead to even better performance in future work.

---

[2]A constant term is assumed to be absorbed in $x$ for clearer exposition.

[3]See the supplement for the exact quantification of the number of weights.

[4]Skip connections such as those seen in residual network-type architectures may end up passing the input to upper layers, but it is unlikely that the input would be consistently passed undisturbed to all upper layers.

## 2 Related Work

There are numerous types of explainable machine learning methods. Given our proposal of an interpretable neural architecture, we focus our discussion of related work on methods that yield global explanations or interpretable models or neural architectures, even though the latter may be opaque.

**Black-Box Neural Architectures.** MLP is a standard neural architecture typically composed of a fully-connected network [15]. However, MLPs have limitations in their representation and performance, leading to many modern architectures. Convolutional neural networks (CNNs) [16] have been very successful in modeling image data. Further improvements were seen with residual network (ResNet) [19] type of architectures which employ ideas such as skip connections. This idea is now used in many other architectures such as DenseNet [22] and MobileNetV2 [43]. Transformers [47] have seen immense success for text data and recently were also shown to perform well for images [14]. $\Pi$-nets [7] were recently shown to perform well for images where they learn a high degree polynomial using tensor decomposition to reduce dimensionality of the search space. Neural networks with activation functions other than the standard ReLU or sigmoid such as those based on Padé approximation [33] have also been suggested. Other architectures such as pi-sigma networks [46] and capsule networks [42] have also found wide appeal.

**Globally Interpretable Models.** Standard machine learning models such as logistic regression, decision trees [5], and rule based models [41, 10] are globally interpretable. Generalized additive models (GAMs) [6] and their more recent variants such as neural additive models (NAMs) [1] and explainable boosted machines (EBMs) [36] also belong to this category. LassoNet [26] is one of the most recently proposed architectures that can be considered interpretable. However, it is restrictive in the sense that if a feature is not selected by itself, it will not appear in any interaction terms. This precludes accurate estimation of functions which have only interaction terms including the very simple bivariate function $x_1 x_2$.

**Local to Global Post Hoc Methods.** There are post hoc explanation methods which take local explanations and create global ones. TreeShap [28] creates a global SHAP explanation for tree based models. While, model agnostic multilevel explanation (MAME) [40] can create global LIME explanations. Alternatively, the global Boolean feature learning (GBFL) [37] method leverages local constrastive explanations [13] to create globally interpretable rule-based models.

**Self-Explaining Models.** Another category of models may not be globally interpretable, but provides local explanations without post hoc mechanisms. [2] is suited for tabular and image data, whereas [53] is suitable for text data. [20] is a framework that provides explanations for new examples if explanations are available for training examples. All of these methods, however, do not readily expose the global behavior of the model.

## 3 Preliminaries

We now introduce some notation and discuss equivalent forms for representing continued fractions. We also discuss some of their properties. As mentioned in the introduction, the generalized form for a continued fraction is $a_0 + \cfrac{b_1}{a_1 + \cfrac{b_2}{a_2 + \cdots}}$, where $a_k$s and $b_k$s can be complex numbers. If none of the $a_k$ or $b_k$ are zero $\forall k \in \mathbb{N}$, then using equivalence transformations [23], one can create simpler equivalent forms where either the $b_k = 1$ or the $a_k = 1 \ \forall k \in \mathbb{N}$, with $a_0 = 0$ in the latter form. A more concise way to write these two forms is as follows: i) $a_0 + \cfrac{1}{a_1 + \cfrac{1}{a_2 + \cdots}} \equiv a_0 + \cfrac{1}{a_1 +} \cfrac{1}{a_2 + \cdots}$ and ii) $\cfrac{b_1}{1 + \cfrac{b_2}{1 + \cdots}} \equiv \cfrac{b_1}{1 +} \cfrac{b_2}{1 + \cdots}$. Form i) is known as *canonical form*. We will interchangeably use the different forms in the paper based on convenience. One of the nice properties of continued fractions is that in representing any real number with natural number parameters $a_k, b_k \in \mathbb{N}$, the rational approximations formed by any of its finite truncations (termed *convergents*) are closer to the true value than any other rational number with the same or smaller denominator. A continued fraction is therefore the "best" possible rational approximation in this precise sense [23, 31].

In the case where $a_k$ and/or $b_k$ are linear *functions* of a variable $x \in \mathbb{R}^p$, these can be written as functions expressible by power series expansions around $x = 0$, where there is a one-to-one correspondence between the coefficients of both [23, 31]. Hence, given parameter vectors $w_k \in \mathbb{R}^p$,

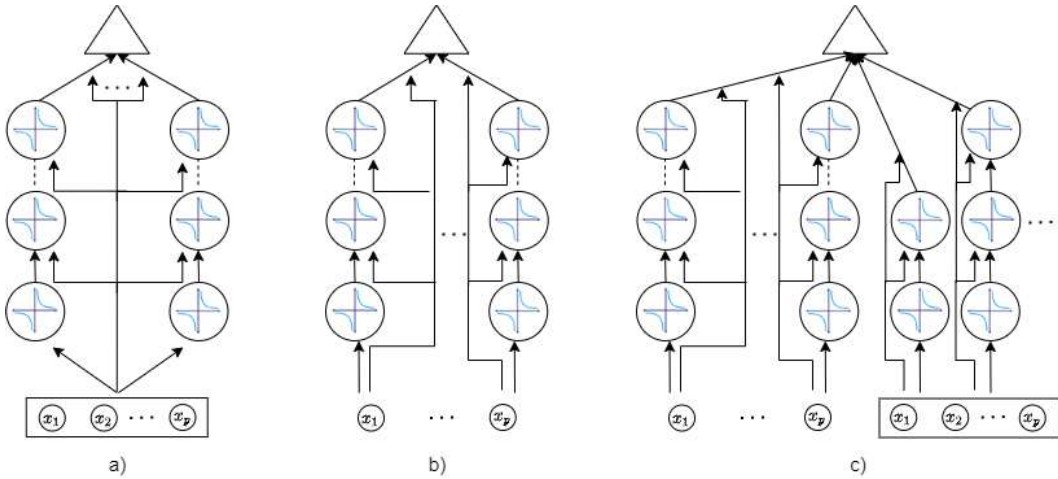

Figure 2: Three variants of the CoFrNet architecture. In all three variants, the output (top triangle) is a linear combination of the ladders below it. a) CoFrNet-F is the full-fledged variant where each ladder receives the whole input $x$ at every stage. b) CoFrNet-D is a diagonalized variant where each ladder only receives one of the input dimensions $x_j$ and hence is an additive model. c) CoFrNet-DL is a combination of the diagonalized variant and the full variant. The full ladders are of increasing depth and can be understood to capture the respective order of interactions.

we can write the following equality as functions of $x$:

$$w_0 x + \frac{1}{w_1 x +} \frac{1}{w_2 x + \cdots} = \sum_{i_1,\ldots,i_p=0}^{\infty} c_{i_1,\ldots,i_p} \prod_{j=1}^{p} x_j^{i_j} \tag{1}$$

for tuples of powers $i_1, \ldots, i_p$ and complex numbers $c_{i_1,\ldots,i_p}$. We will leverage this relationship in Section 4 as one of the strategies to interpret CoFrNets, since it allows us to express a CF as a power series and hence derive feature attributions for individual features as well as their interactions.

## 4   CoFrNet Architecture

In this section, we present the proposed architecture based on CFs with three variants. We then discuss aspects of effectively training such architectures and how to obtain feature attributions for interpretation.

**Proposed Architecture.** We focus on continued fractions in canonical form, with unit numerators $b_k = 1$. As in (1), we let the denominators $a_k = w_k^T x$ be linear functions of the input $x$. Then with $d$ denoting the depth, the basic continued-fractional function that we work with is

$$f(x; w) = a_0 + \frac{1}{a_1 +} \cdots \frac{1}{a_{d-1} +} \frac{1}{a_d}, \quad a_k = w_k^T x. \tag{2}$$

Each such function corresponds to the diagram in Figure 1. We refer to such a function as a "ladder" due to this pictorial representation with a rail and rungs that carry the input to each node.

We propose three variants of the architecture, shown in Figure 2, where each variant is a linear combination of functions $f$ in (2), i.e., a combination of ladders. We propose CoFrNet-F as a full-fledged variant in which all ladders receive the full input $x$ at each layer. We propose a diagonalized variant CoFrNet-D, in which each ladder operates on a single input dimension $x_j$, i.e., $f_j(x_j; w^{(j)})$. The linear combination of these ladders is therefore an additive model and directly interpretable [18]. Finally, we propose CoFrNet-DL, which contains both single-feature ladders and full ladders of increasing depth starting at depth two and hence capturing interactions to that order. This variant combines the benefits of the other two architecture variants.

**Training Strategies.** As discussed in the introduction, we regard the proposed architectures as neural architectures in which the input is passed to all layers, linear functions with weights $w_k$ are computed,

and the nonlinear activation function is the reciprocal $z \mapsto 1/z$. All variants are differentiable and can thus be trained using standard techniques such as ADMM and backpropagation and popular frameworks such as TensorFlow or PyTorch. Other commonly used ideas such as dropout [44] could also be leveraged for better generalization.

The most natural choice is to jointly train all the ladders; however, one could envision other training strategies. For example, one could use boosting where one or a group of ladders are first (jointly) trained and where we successively train new ladders on the residuals with appropriate example weighting. One could also incrementally fit each layer within a ladder. Another strategy might be to collapse the linear combination of ladders into a single rational function. However, it may be a challenge to tie the coefficients of this rational function to the weights $w_k$; not constraining the rational function coefficients in this way would result in an exponential number of coefficients to estimate (exponential in the depth of the ladders).

**Handling Poles.** A key issue that arises from the $\frac{1}{z}$ reciprocal nonlinearity is that the denominator may go to zero during training, leading to the function being undefined at that point (a *pole* in the context of rational functions). To tackle this issue, we slightly alter the activation function to $\mathsf{sgn}(z)\frac{1}{\max(|z|,\epsilon)}$ for some (small) $\epsilon > 0$, where $|\cdot|$ denotes absolute value. The $\epsilon$ can be fixed to a small positive value or tuned during training. Other solutions to this problem involve either restricting each of the denominators to be positive [33] or the final denominator of the rational function to be positive [3]. Both of these constraints can be restrictive as well as computationally challenging.

**Interpretability.** We now discuss two strategies to interpret the full-fledged version of our architecture CoFrNet-F. Both strategies exploit its functional form. As mentioned, CoFrNet-D is an additive model and can be interpreted by visualizing the univariate functions $f_j(x_j; w^{(j)})$ that compose it.

**i) Interpretation using Continuants (IC):** It is well-known from the theory of continued fractions [23] that $f(x; w)$ in (2) can be expressed as the following ratio of polynomials,

$$f(x; w) = a_0 + \cfrac{1}{a_1+} \cdots \cfrac{1}{a_{d-1}+} \cfrac{1}{a_d} = \frac{K_{d+1}(a_0, \ldots, a_d)}{K_d(a_1, \ldots, a_d)}, \tag{3}$$

where the polynomials $K_k$, known as *continuants*, satisfy the recursion

$$K_0 = 1, \qquad K_1(a_d) = a_d, \tag{4}$$

$$K_k(a_{d-k+1}, \ldots, a_d) = a_{d-k+1} K_{k-1}(a_{d-k+2}, \ldots, a_d) + K_{k-2}(a_{d-k+3}, \ldots, a_d). \tag{5}$$

The following result (proven in the supplement) provides a compact expression for the gradient of $f(x; w)$ with respect to the inputs $x_j, j = 1, \ldots, p$.

**Proposition 1.** *The partial derivative of $f(x; w)$ with respect to $x_j$ is given by*

$$\frac{\partial f(x; w)}{\partial x_j} = \sum_{k=0}^{d} (-1)^k \left( \frac{K_{d-k}(a_{k+1}, \ldots, a_d)}{K_d(a_1, \ldots, a_d)} \right)^2 w_{jk}.$$

Proposition 1 provides a computationally efficient means to compute the gradient of a ladder with respect to its inputs, which is useful for multiple feature-based methods of interpretation [34]. Given an input $x$, and assuming that the linear functions $a_k = w_k^T x$ have already been computed in evaluating $f(x; w)$, the continuants $K_k$ can be computed using (5) in $O(d)$ operations. Then for each input $x_j$, the sum in Proposition 1 also requires $O(d)$ operations, for a total of $O(dp)$ operations for all $x_j$. Proposition 1 additionally suggests an interpretation of the coefficients $w_{jk}$ as contributions to the partial derivative for $x_j$, weighted by ratios of continuants and with alternating signs.

For linear combinations of ladders $f$ as in Figure 2, the above result extends straightforwardly since differentiation is a linear operation. This yields feature attributions in $O(Ldp)$ time for $L$ ladders.

**ii) Interpretation using Power Series (IPS):** The above method using continuants gives first-order attributions at a per example level. To obtain higher-order as well as first-order global attributions, we turn to the representation of a ladder in (1) as a multivariate power series, where as mentioned before there is a one-to-one mapping between the coefficients of the two forms. A linear combination of ladders, which our architecture entails, can also be represented by a multivariate power series by summing the coefficients $c_{i_1, \ldots, i_p}$ for each monomial term. These coefficient sums thus provide attributions for individual features $x_j$ as well as higher-order interactions, up to the depth of the ladders.

For low-depth ladders, it is possible to manually equate and find the appropriate coefficients based on a linear recurrence relation [39]. However in general, manual computation can be too laborious. For such cases we recommend using symbolic manipulation tools such as Mathematica [51]. For a function $g$ that is a linear combination of ladders $f$ of depth $d$, one can obtain the power series expansion up to order $dp$ by applying the following set of Mathematica operations:

$$\text{N}\left[\text{Normal}\left[\text{Series}\left[g, \{x_1, 0, d\}, \cdots, \{x_p, 0, d\}\right]\right]\right] \tag{6}$$

where "Series" produces a Taylor series expansion, "Normal" implies normalized form, and "N" represents fractional coefficients as decimals. The appropriate coefficients can then be picked off to determine feature attributions or attributions for interactions.

## 5  Universal Approximation

We now prove (Theorem 2) that a linear combination of continued fractions has the property of universal approximation. More precisely, we show this for the family of functions that are linear combinations of a finite number of continued fractions, each with finite depth and linear layers. Our strategy essentially comprises three steps: i) showing polynomials of linear functionals ($\mathcal{PL}$) are a unital subalgebra [4] and separating on the domain, ii) applying the Stone-Weierstrass theorem [45] to show that they are thus dense in the space of bounded continuous functions, and iii) showing that $\mathcal{PL}$ are a subset of the aforementioned class of functions, i.e. finite number of finite-depth continued fractions where each layer is a linear function of the input. This implies the latter class is also dense in the space of bounded continuous functions, a.k.a. universal approximators.

Without loss of generality we consider the domain $\chi = [0, 1]^p$ along with the usual Euclidean metric $d(x, y) = \|x - y\|_2$, $x, y \in \chi$. Since $\chi$ is bounded and closed, it is a compact metric space. The space of bounded continuous functions $C(\chi, \mathbb{R}) = \{f : \chi \mapsto \mathbb{R} : f \text{ is continuous, } \exists M \text{ s.t. } |f(x)| \leq M \ \forall x \in \chi\}$. Let $\|f(x)\|_\infty = \max_{x \in \chi} |f(x)|$. Also for the proof we explicitly mention the constant term for each linear function which was subsumed in $x$ in previous sections.

**Definition 1.** *(Identity Function) Define* $\text{Id}(x)$ *to be the identity function where* $\text{Id}(x) = 1 \ \forall x \in \chi$.

**Definition 2.** $\mathcal{P} \subset C(\chi, \mathbb{R})$ *is a unital subalgebra if a)* $\text{Id}(x) \in \mathcal{P}$, *b)* $\forall f, g \in \mathcal{P}, f * g \in \mathcal{P}$, *c)* $\forall f, g \in \mathcal{P}$ *and* $\alpha, \beta \in \mathbb{R}, \alpha f + \beta g \in \mathcal{P}$.

**Definition 3.** $\mathcal{P} \subset C(\chi, \mathbb{R})$ *is separating on the domain* $\chi$ *if* $\forall x, y \in \chi, \ x \neq y, \exists f \in \mathcal{P} : f(x) \neq f(y)$.

**Theorem 1.** *(Stone-Weierstrass Theorem [45]) If* $\chi$ *is a compact metric space, and if* $\mathcal{P} \subset C(\chi, \mathbb{R})$ *is a unital subalgebra and separating on the domain* $\chi$, *then* $\mathcal{P}$ *is dense in* $C(\chi, \mathbb{R})$ *with respect to the* $\ell_\infty$ *metric.*

**Definition 4.** *(Polynomials of Linear Functions) Define*

$$\mathcal{PL} = \left\{ c_0 + \sum_{S \in \mathcal{S}} c_S \prod_{k \in S} (u_k^T x) : c_0, c_S \in \mathbb{R}, \ u_k \in \mathbb{R}^p \ \forall k \in [m], \ \mathcal{S} \subset 2^{[m]}, \ m \in \mathbb{N} \right\} \tag{7}$$

*to be the set of polynomials on linear functions* $u_k^T x$ *of* $x$.

In the above definition, note that we may have $u_l = u_k$ for $l \neq k$ to obtain higher powers of $u_k^T x$.

**Lemma 1.** $\mathcal{PL}$ *is a unital subalgebra and is separating on* $\chi$. $\mathcal{PL}$ *is dense in* $C(\chi, \mathbb{R})$.

*Proof.* Let $f(x), g(x) \in \mathcal{PL}$. It is easy to see that $f(x) * g(x) \in \mathcal{PL}$ and $\alpha f(x) + \beta g(x) \in \mathcal{PL}$ for all $\alpha, \beta \in \mathbb{R}$. Further, setting $c_0 = 1$ and $c_S = 0$ in the definition of $\mathcal{PL}$ yields the identity function. Therefore, $\mathcal{PL}$ is a unital subalgebra.

For any two $x \neq y$, $x, y \in \chi$, let $u \in \mathbb{R}^p$, $u \neq 0$ be such that it does not belong to the null space of $x - y$. Such a $u$ can always be found by the rank-nullity theorem applied to the subspace $\text{span}\{x - y\}$ in the vector space $\mathbb{R}^p$. Now, consider $f(s) = c_0 + u^T s \in \mathcal{PL}$. Then, $f(x) \neq f(y)$ since $u^T(x - y) \neq 0$. This shows that $\mathcal{PL}$ is separating in the domain $\chi$. Hence, by Theorem 1, $\mathcal{PL}$ is dense in $C(\chi, \mathbb{R})$. $\qquad\square$

**Definition 5.** *(Continued Fractions with Linear Functions) Let*

$$\mathcal{CFL} = \left\{ \frac{v_0^T x + \alpha_0}{1+} \frac{v_1^T x + \alpha_1}{1 + w_1^T x + \beta_1+} \frac{v_2^T x + \alpha_2}{1 + w_2^T x + \beta_2+} \cdots \frac{v_d^T x + \alpha_d}{1 + w_d^T x + \beta_d} : \right.$$

$$\left. v_0, v_k, w_k \in \mathbb{R}^p, \ \alpha_0, \alpha_k, \beta_k \in \mathbb{R}, \ 1 \leq k \leq d, \ d \in \mathbb{N} \right\} \tag{8}$$

*be the set of finite-depth continued fractions with affine functions $v_k^T x + \alpha_k$ and $w_k^T x + \beta_k$ as numerators and denominators.*

In the above definition, we are explicitly writing the constant terms $\alpha_k, \beta_k$ for clarity.

Given a set $A$ of functions on $\chi$, let $A \bigoplus A = \{\alpha a + \beta b : a, b \in A, \alpha, \beta \in \mathbb{R}\}$ be the set of linear combinations of two functions from $A$, and $\bigoplus^L A$ the set of linear combinations of $L$ functions from $A$.

**Theorem 2.** *(Representation Theorem)* $\mathcal{PL} \subset \bigcup_{L=1}^{\infty} \bigoplus^L \mathcal{CFL}$. *Also,* $\bigcup_{L=1}^{\infty} \bigoplus^L \mathcal{CFL}$ *is dense in* $C(\chi, \mathbb{R})$.

*Proof.* By Euler's formula for continued fractions we have:

$$\frac{a_0}{1+} \frac{-a_1}{1 + a_1+} \frac{-a_2}{1 + a_2+} \cdots \frac{-a_d}{1 + a_d} = a_0 + a_0 a_1 + \ldots a_0 a_1 \ldots a_d. \tag{9}$$

By applying the above formula twice to two nested sums, we have:

$$a_0 a_1 \ldots a_d = \left[ \frac{a_0}{1+} \frac{-a_1}{1 + a_1+} \frac{-a_2}{1 + a_2+} \cdots \frac{-a_d}{1 + a_d} \right] - \left[ \frac{a_0}{1+} \frac{-a_1}{1 + a_1+} \frac{-a_2}{1 + a_2+} \cdots \frac{-a_{d-1}}{1 + a_{d-1}} \right]. \tag{10}$$

Now to represent a monomial $c \prod_{k \in [d]} (u_k^T x)$, $c \in \mathbb{R}$, we observe that we need to set $a_0 = c$, $a_k = (u_k^T x) \ \forall k \in \{1, ..., d\}$ in (10). This in turn can be realized as a member of $\mathcal{CFL}$ by setting $v_0 = 0$, $\alpha_0 = c$, $w_k = -v_k = u_k$, and $\alpha_k = \beta_k = 0$ for $k = 1, \ldots, d$. Hence we have $c \prod_{k \in [d]} (u_k^T x) \in \mathcal{CFL} \bigoplus \mathcal{CFL}$.

Now consider $f(x) = c_0 + \sum_{S \in \mathcal{S}} c_S \prod_{k \in S} (u_k^T x) \in \mathcal{PL}$. Then we have $f \in \bigoplus^{2|\mathcal{S}|+1} \mathcal{CFL}$ by doing a term by term expansion in terms of $\mathcal{CFL} \bigoplus \mathcal{CFL}$. This implies that $\mathcal{PL} \subset \bigcup_{L=1}^{\infty} \bigoplus^L \mathcal{CFL}$.

Combining with Lemma 1 implies that $\bigcup_{L=1}^{\infty} \bigoplus^L \mathcal{CFL}$ is dense in $C(\chi, \mathbb{R})$. $\square$

**Remark.** Note that $\bigcup_{L=1}^{\infty} \bigoplus^L \mathcal{CFL}$ does contain functions with singularities in $\mathbb{R}^p$. This implies that it contains functions that are not in $C(\chi, \mathbb{R})$. But nevertheless, it is dense in $C(\chi, \mathbb{R})$. The presence of singularities is the reason why it is not possible to directly apply proof techniques using the Hahn-Banach theorem [9], which are used for proving representation theorems for two-layer neural networks. It is noteworthy that one only needs linear functions in every layer of a continued fraction and linear combinations of these to represent any bounded function on a compact set.

**Compactness of Representation for Learning Sparse Polynomials.** If one wants to learn a sparse polynomial in the variables $x_j$ where the number of non-zero monomials $|\mathcal{S}|$ is a constant and degree bounded, i.e. $|S| \leq d$, $\forall S \in \mathcal{S}$, Lasso-based techniques would require a representation which is $p^d$ in size (although sample complexity may be polynomial in $d \log p$) [35]. However, our representation would require only $2|\mathcal{S}| + 1$ parameterized ladders each of depth at most $d$. Efficient learning of sparse polynomials using such compact representations is an interesting direction for future work.

# 6 Experiments

We now conduct synthetic and real data experiments. The goal of the synthetic experiments is two fold: i) to show that we can accurately model well known non-linear functions [50] (viz. Matyas

function, Rosenbrock function, etc.), but more importantly ii) to show that our architecture lends itself to global interpretation using the two strategies described in Section 4. Performance comparisons in terms of mean absolute percentage error (MAPE) with MLPs of the same depth and with similar number of parameters are in the supplement, as our main intent here is to showcase interpretability.

We then experiment on seven real public datasets covering tabular, text and image data. For six of them in addition to MLP we compare against four well known interpretable models namely, GAMs (`github.com/dswah/pyGAM`), NAMs (`github.com/nickfrosst/neural_additive_models`), EBMs (`github.com/interpretml/interpret`), and CART decision trees as well as the recent LassoNet (`github.com/lasso-net/lassonet`) with the main intent being to compare test accuracies. For the text data we used Glove embeddings [38]. We report feature attributions for some of the datasets in the main paper.

We average results over five random train/validation/test splits (65%/5%/30%) for all datasets except those that come with their own pre-specified test set. A two P100 GPU system with 120 GB RAM was used to run the experiments. In the activation function, $\epsilon$ was set to 0.1 to mitigate poles.

## 6.1 Synthetic Experiments

In these experiments we use the CoFrNet-F variant to model different synthetic functions, based on a sample of 300 points for each function. We compute feature attributions using the continuants strategy, and the entire function using the power series strategy mentioned in Section 4. A single full ladder is used in each case and the depth is equal to the degree of the function we are approximating. For non-polynomial functions we set the depth to be six. We choose CoFrNet-F since it is the hardest variant to interpret; we show that it can be interpreted.

Figure 3 shows two well-known non-linear functions: the Matyas function and the Rosenbrock function. CoFrNet-F is able to accurately approximate both (7.31% and 13.08% MAPE respectively). Importantly, the IPS interpretation leveraging the one-to-one correspondence to power series is able to replicate the functions quite closely. The constructed interpretation is not only close in prediction, but also in (univariate) feature attribution. For Matyas, we even recover higher order coefficients accurately (possibly due to a better fit). The linear and constant terms have very small coefficients in the Matyas function approximation, making the IPS and original function very similar. Moreover, the feature attributions for the linear terms are also recovered by IC, the closed-form formula involving continuants.

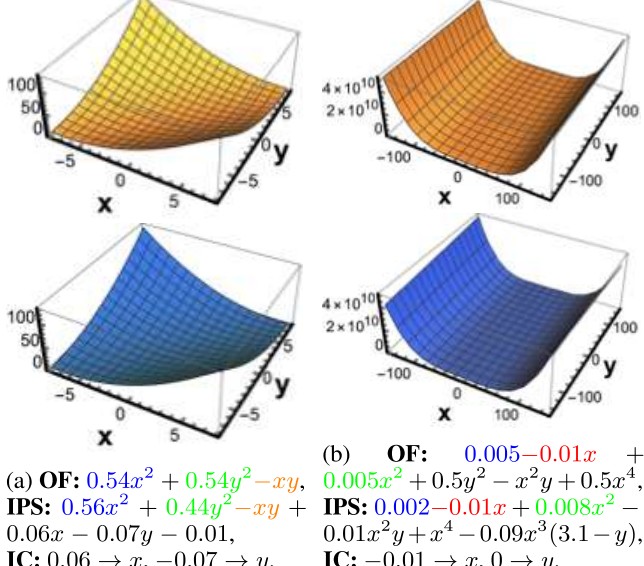

(a) **OF:** $0.54x^2 + 0.54y^2 - xy$, **IPS:** $0.56x^2 + 0.44y^2 - xy + 0.06x - 0.07y - 0.01$, **IC:** $0.06 \rightarrow x, -0.07 \rightarrow y$.

(b) **OF:** $0.005 - 0.01x + 0.005x^2 + 0.5y^2 - x^2y + 0.5x^4$, **IPS:** $0.002 - 0.01x + 0.008x^2 - 0.01x^2y + x^4 - 0.09x^3(3.1 - y)$, **IC:** $-0.01 \rightarrow x, 0 \rightarrow y$.

Figure 3: Original function (OF; in yellow) and the corresponding IPS approximation (in blue) for the Matyas function (left) and Rosenbrock function (right). The subfigure caption lists the OF, IPS, and feature attributions from IC. The equations for OF and IPS are normalized by maximum coefficient. Coefficients of the same order terms (that are close) are color-coded for ease of comparison. CoFrNet-F is able to approximate the shape of the functions well. The (univariate) feature attributions for IPS and IC are consistent.

## 6.2 Real Data Experiments

We evaluate our approach relative to other approaches on Credit Card, Magic and Waveform tabular datasets [12], the sentiment analysis [29] and Quora Insincere Questions [24] text datasets, and the CIFAR10 [25] image dataset. We also experimented with our approach on the ImageNet dataset [11]. The dataset characteristics are in the supplement. We report performance of the CoFrNet-DL architecture, which was the best performing among the three variants and also

Table 1: Test accuracies of different methods on six real datasets. For datasets without pre-specified test sets, a paired t-test was conducted to determine statistical significance. DNC denotes 'did not converge.' Best results (within statistical error and excluding SOTA) are in bold.

| Methods | Interpretable | Waveform | Magic | Credit Card | CIFAR10 | Sentiment | Quora |
|---|---|---|---|---|---|---|---|
| CoFrNet-DL | Yes | 0.81 | **0.84** | 0.69 | **0.87** | **0.84** | **0.88** |
| CoFrNet-D | Yes | 0.69 | 0.76 | 0.66 | 0.38 | 0.80 | 0.75 |
| GAM | Yes | **0.85** | **0.85** | **0.72** | DNC | 0.51 | DNC |
| NAM | Yes | **0.86** | 0.81 | 0.69 | 0.38 | 0.50 | 0.49 |
| EBM | Yes | **0.85** | **0.85** | **0.72** | 0.40 | 0.59 | 0.49 |
| CART | Yes | 0.75 | 0.79 | 0.69 | 0.29 | 0.52 | 0.73 |
| LassoNet | Yes | 0.84 | 0.76 | 0.67 | 0.28 | 0.50 | 0.53 |
| MLP | No | 0.34 | 0.65 | 0.50 | 0.35 | **0.83** | 0.85 |
| SOTA | No | 0.86 | 0.86 | 0.75 | 0.99 | 0.96 | 0.94 |

CoFrNet-D to show how well our simplest form performs. For CoFrNet-D, the depth of the univariate ladders is varied up to 250. For CoFrNet-DL, besides the $p$ univariate ladders, we consider up to 50 full ladders with increasing depth (maximum depth of 50) as per the architecture in Figure 2. We used early stopping, dropout, batching and Adam optimizer with weight decay. For CIFAR10 we also did data augmentation (i.e. random cropping and flipping). The best performance for most datasets using CoFrNet-DL was obtained using ladders of depth 12 or less.

We observe in Table 1, that the CoFrNet-DL model is competitive with other interpretable models on the tabular datasets (i.e. within few percent) and within 6% of the state-of-the-art (SOTA) black-box models for these datasets (gradient boosted trees) [37]. On images and text, although we are farther off from SOTA black-box models ([17] for CIFAR10 and [52] for text), we are significantly better than other interpretable models, and similar or better than (uninterpretable) MLP. We believe this is because CoFrNets can compactly represent a rich class of functions in high-dimensional space where properties of CFs such as fast convergence are likely witnessed. We additionally also trained the CoFrNet-DL model on ImageNet and obtained an accuracy of 0.69, which is comparable to ResNet-18 type architectures.

We showcase the interpretability of our CoFrNet-DL model in Figure 4. Figures 4a, 4b and 4c depict the (functional) behavior of the most important diagonalized ladders in our CoFrNet-DL models trained on the Waveform, Magic and Credit Card datasets respectively. Similar plots for the top three features in each dataset are provided in the supplement. The important features also seem to make semantic sense given the respective tasks. For example, in the Credit Card dataset "Bill statement in April, 2005" should have an impact on predicting payment defaults, since someone not having repaid their previous balance would presumably have a higher likelihood of defaulting. Figure 4d, shows the feature attributions for individual images using the IC strategy. In this case global attributions make little sense to report, which is why we report local explanations. Our explanations seem to focus on critical aspects such as wings of the plane, the body and face of the horse, and the face and ears of the dog. More such explanations are again provided in the supplement. Figures 4e and 4f depict the most important words, filtering out articles, prepositions and auxiliary verbs, that are highlighted by our CoFrNet-DL models on the Sentiment and Quora datasets. It makes sense that words such as "good", "bad" and "like" should play an important role in gauging sentiment, while words such as "some", "people" especially taken together, could indicate sarcasm/insincerity. More discussion of these results is provided in the supplement.

# 7   Discussion

In this paper, we have proposed CoFrNets, a new neural architecture inspired by continued fractions. We have theoretically shown its universal approximation ability, empirically shown its competence on real-world datasets where we are either competitive or much better than other interpretable models as well as MLPs, and analytically shown how to tease interpretability out from it. The training optimization relies on specific properties of CFs: while CFs are rational functions, optimizing rational functions directly leads to either exponentially-many coefficients or constraints that are difficult to enforce, but these pitfalls do not happen when the CF representation is used. Similarly, interpretation is obtained efficiently and naturally by leveraging the theory of CFs.

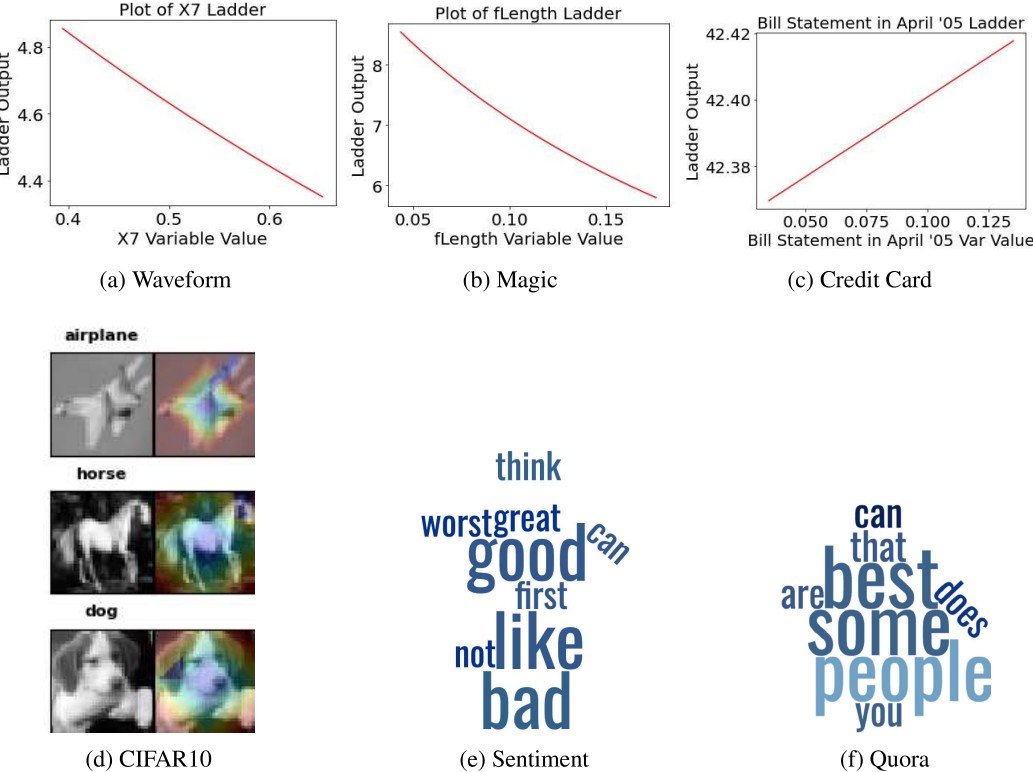

Figure 4: In the first row for the three tabular datasets (a-c), we see the behavior of the most important features as indicated by our CoFrNet-DL model. In the second row, we first see (d) local explanations for CIFAR10 using the IC strategy. Shades closer to blue indicate high importance, while those closer to red imply low importance (colormap in the supplement). The next two figures (e, f) are word clouds highlighting the words our CoFrNet-DL model considers as most important. More example explanations are provided in the supplement.

We hypothesize that CoFrNets have favorable adversarial robustness properties due to their functional simplicity and we intend to test this in future work. The hypothesis arises from the following argument. Ladders of small depth, which we find to have good accuracy, yield smooth low-degree rational functions that make it difficult for small input perturbations to produce large changes in output. We may even be able to analytically compute adversarial robustness metrics akin to the CLEVER score for CoFrNets [48].

Beyond this initial proposal of a new architecture, we admit there is room for improvement to achieve empirical accuracies that match or outdo state-of-the-art black-box models across modalities. Since CoFrNet can be viewed as a neural architecture, we have been able to exploit the well-developed tools available to train neural architectures. It is possible however that different training strategies such as those mentioned in Section 4 could be advantageous: each ladder could be built rung by rung, or a linear combination of ladders could be built incrementally. Similarly, while the $\epsilon$ modification of the $1/z$ function is a practical solution to avoid singularities, it is possible that it also limits the expressiveness of the function, and more advanced ways to defeat poles [3] could be less restrictive. In addition, known tactics from other neural architectures (convolutional blocks, pooling, maxout) or even something new may be warranted. It is possible that convolutional blocks may have a natural implementation based on older filter design literature in signal processing that builds upon CFs [32].

## Funding and Conflicts of Interest

All authors were employed by IBM Corporation when this work was conducted. There were no other sources of funding.

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
