

Figure 5: Above (`https://en.wikipedia.org/wiki/Coffer`) we see an example of a coffer in building architecture, which is a series of (square) sunken panels.

Table 2: Public dataset characteristics, where $N$ denotes dataset size and $p$ is the dimensionality. For the last two text datasets $p$ is based on our glove embedding.

| Dataset | Modality | $N$ | $p$ | # of Classes |
|---|---|---|---|---|
| Credit Card | Tabular | 30K | 24 | 2 |
| Magic | Tabular | 19020 | 11 | 2 |
| Waveform | Tabular | 5K | 40 | 3 |
| CIFAR-10 | Image | 60K | $32 \times 32$ | 10 |
| Sentiment | Text | 50K | 12.5K | 2 |
| Quora | Text | 99933 | 4K | 2 |
| ImageNet | Image | 14M | $224 \times 224$ | 1000 |

## A  (Additional) Real Data Details

Table 2 shows the real dataset characteristics. For the Sentiment dataset each word had a 50 dimensional embedding where the (max) sentence length was set to 250 making the dimensionality of a particular input to be 12,500. For Quora, each word had a 20 dimensional embedding and the (max) sentence length was set to 200 making the dimensionality of a particular input to be 4,000.

The top three attributions for Waveform were X7, X11 and X33. For Quora they were "some", "people" and "best". Interestingly, the least important words in Quora were inquisitive verbs such as "Why", "What", "How", "Can", "If" and "Which". This is understandable as those are present in (almost) every question (or input) and are thus not helpful in distinguishing insincere questions from actual ones.

## B  MLP vs CoFrNet-F on Synthetic Functions

We now compare the performance of MLPs to CoFrNet-F on well known synthetic functions given in Table 3. We consider single ladder CoFrNet-F, whose depth we set to be equal to the degree of the function if it is a polynomial, else we set it to six. For a fair comparison we also set the depth of the MLP to be the same as our architecture. The width for the MLP is then set so that the number of parameters in it are as similar to ours as possible. To do this we state the following simple formulas that connect depth, width and number of parameters.

If $p$ is the dimensionality of the input, $q$ the dimensionality of the output, $d$ the depth and $L$ the width (i.e. number of hidden nodes for MLP or number of ladders for CoFrNet) then,

Table 3: Below we see the (un-normalized) functional form of 10 different functions that we perform (synthetic) experiments on [50].

| Function | Formula |
|---|---|
| Beale | $(1.5 - x + xy)^2 + (2.25 - x + xy^2)^2 + (2.625 - x + xy^3)^2$ |
| Goldstein–Price | $(1 + (x + y + 1)^2(19 - 14x + 3x^2 - 14y + 6xy + 3y^2)) \times$ $(30 + (2x - 3y)^2(18 - 32x + 12x^2 + 48y - 36xy + 27y^2))$ |
| Booth | $(x + 2y - 7)^2 + (2x + y - 5)^2$ |
| Cross In Tray | $-.0001(|\sin(x)\sin(y)\exp(|100 - \frac{\sqrt{x^2+y^2}}{\pi}|)| + 1)^{0.1}$ |
| Three Hump Camel | $2x^2 - 1.05x^4 + \frac{x^6}{6} + xy + y^2$ |
| Himmelblau | $(x^2 + y - 11)^2 + (x + y^2 - 7)^2$ |
| Bukin N6 | $100\sqrt{|y - .01x^2|} + .01|x + 10|$ |
| Matya's | $.26(x^2 + y^2) - .48xy$ |
| Levi N13 | $\sin^2(3\pi x) + (x - 1)^2(1 + \sin^2(3\pi y)) + (y - 1)^2(1 + \sin^2(2\pi y))$ |
| Rosenbrock | $(1 - x)^2 + 100(y - x^2)^2$ |

Table 4: Below we see the mean absolute percentage error (MAPE), where the percentage is with respect to the $\max - \min$ range of function values amongst the sampled examples, of (single ladder) CoFrNet-F and MLP with same depth and with similar number of parameters on the 10 well known synthetic functions. Best results are bolded.

| Function | CoFrNet-F | MLP |
|---|---|---|
| Beale | **16.512** | 19.480 |
| Goldstein–Price | **12.045** | 20.966 |
| Booth | **11.885** | 21.932 |
| Cross In Tray | 9.926 | **5.591** |
| Three Hump Camel | **36.250** | 37.315 |
| Himmelblau | **25.545** | 34.420 |
| Bukin N6 | **20.073** | 40.795 |
| Matya's | **7.311** | 15.525 |
| Levi N13 | **24.873** | 28.751 |
| Rosenbrock | **13.080** | 44.520 |

CoFrNet-F: # of Parameters $= pL(d - 1) + Lq$, MLP: # of Parameters $= pL + (d - 2)L^2 + Lq$

We see in Table 4, that we outperform MLP in majority of the cases showcasing the power of our architecture.

## C  Proof of Proposition 1

*Proof.* The proposition follows from the chain rule and Lemma 2 below:

$$\frac{\partial f(x; w)}{\partial x_j} = \sum_{k=0}^{d} \frac{\partial}{\partial a_k} \frac{K_{d+1}(a_0, \ldots, a_d)}{K_d(a_1, \ldots, a_d)} \frac{\partial a_k}{\partial x_j} = \sum_{k=0}^{d} \frac{\partial}{\partial a_k} \frac{K_{d+1}(a_0, \ldots, a_d)}{K_d(a_1, \ldots, a_d)} w_{jk}.$$

□

**Lemma 2.** *We have*

$$\frac{\partial}{\partial a_k} \frac{K_{d+1}(a_0, \ldots, a_d)}{K_d(a_1, \ldots, a_d)} = (-1)^k \left( \frac{K_{d-k}(a_{k+1}, \ldots, a_d)}{K_d(a_1, \ldots, a_d)} \right)^2.$$

*Proof.* To compute the partial derivative of the ratio of continuants above, we first determine the partial derivative of a single continuant $K_k(a_1, \ldots, a_k)$ with respect to $a_l$, $l = 1, \ldots, k$. We use the

representation of $K_k$ as the determinant of the following tridiagonal matrix:

$$K_k(a_1, \ldots, a_k) = \det \begin{bmatrix} a_1 & 1 & & \\ -1 & a_2 & \ddots & \\ & \ddots & \ddots & 1 \\ & & -1 & a_k \end{bmatrix}. \tag{11}$$

The partial derivatives of a determinant with respect to the matrix entries are given by the *cofactor* matrix:

$$\frac{\partial \det A}{\partial A_{ij}} = \mathrm{co}(A)_{ij},$$

where $\mathrm{co}(A)_{ij} = (-1)^{i+j} M_{ij}$ and $M_{ij}$ is the $(i,j)$-minor of $A$. In the present case, with $A$ as the matrix in (11), we require partial derivatives with respect to the diagonal entries. Hence

$$\frac{\partial K_k(a_1, \ldots, a_k)}{\partial a_l} = M_{ll}.$$

In deleting the $l$th row and column from $A$ to compute $M_{ll}$, we obtain a block-diagonal matrix where the two blocks are tridiagonal and correspond to $a_1, \ldots, a_{l-1}$ and $a_{l+1}, \ldots, a_k$. Applying (11) to these blocks thus yields

$$\frac{\partial K_k(a_1, \ldots, a_k)}{\partial a_l} = K_{l-1}(a_1, \ldots, a_{l-1}) K_{k-l}(a_{l+1}, \ldots, a_k). \tag{12}$$

Returning to the ratio of continuants in the lemma, we use the quotient rule for differentiation and (12) to obtain

$$\begin{aligned} \frac{\partial}{\partial a_k} \frac{K_{d+1}(a_0, \ldots, a_d)}{K_d(a_1, \ldots, a_d)} &= \frac{1}{K_d(a_1, \ldots, a_d)^2} \left( \frac{\partial K_{d+1}(a_0, \ldots, a_d)}{\partial a_k} K_d(a_1, \ldots, a_d) \right. \\ &\qquad \left. - K_{d+1}(a_0, \ldots, a_d) \frac{\partial K_d(a_1, \ldots, a_d)}{\partial a_k} \right) \\ &= \frac{K_{d-k}(a_{k+1}, \ldots, a_d)}{K_d(a_1, \ldots, a_d)^2} \left( K_k(a_0, \ldots, a_{k-1}) K_d(a_1, \ldots, a_d) \right. \\ &\qquad \left. - K_{d+1}(a_0, \ldots, a_d) K_{k-1}(a_1, \ldots, a_{k-1}) \right). \end{aligned} \tag{13}$$

We focus on the quantity

$$K_k(a_0, \ldots, a_{k-1}) K_d(a_1, \ldots, a_d) - K_{k-1}(a_1, \ldots, a_{k-1}) K_{d+1}(a_0, \ldots, a_d) \tag{14}$$

in (13). For $k = 0$ (and taking $K_{-1} = 0$), this reduces to $K_d(a_1, \ldots, a_d)$. Equation (13) then gives

$$\frac{\partial}{\partial a_0} \frac{K_{d+1}(a_0, \ldots, a_d)}{K_d(a_1, \ldots, a_d)} = \left( \frac{K_d(a_1, \ldots, a_d)}{K_d(a_1, \ldots, a_d)} \right)^2 = 1,$$

in agreement with the fact that $a_0$ appears only as the leading term in (3). For $k = 1$, (14) becomes

$$a_0 K_d(a_1, \ldots, a_d) - K_{d+1}(a_0, \ldots, a_d) = -K_{d-1}(a_2, \ldots, a_d)$$

using (5), and hence

$$\frac{\partial}{\partial a_1} \frac{K_{d+1}(a_0, \ldots, a_d)}{K_d(a_1, \ldots, a_d)} = - \left( \frac{K_{d-1}(a_2, \ldots, a_d)}{K_d(a_1, \ldots, a_d)} \right)^2.$$

We generalize from the cases $k = 0$ and $k = 1$ with the following lemma.

**Lemma 3.** *The following identity holds:*

$$\begin{aligned} K_k(a_0, \ldots, a_{k-1}) K_d(a_1, \ldots, a_d) - K_{k-1}(a_1, \ldots, a_{k-1}) K_{d+1}(a_0, \ldots, a_d) \\ = (-1)^k K_{d-k}(a_{k+1}, \ldots, a_d). \end{aligned}$$

Combining (13) and Lemma 3 completes the proof. $\qquad\square$

*Proof of Lemma 3.* We prove the lemma by induction. The base cases $k = 0$ and $k = 1$ were shown above and hold moreover for any depth $d$ and any sequence $a_0, \ldots, a_d$. Assume then that the lemma is true for some $k$, any $d$, and any $a_0, \ldots, a_d$. For $k + 1$, we use recursion (5) to obtain

$$
\begin{aligned}
&K_{k+1}(a_0, \ldots, a_k)K_d(a_1, \ldots, a_d) - K_k(a_1, \ldots, a_k)K_{d+1}(a_0, \ldots, a_d) \\
&= \big(a_0 K_k(a_1, \ldots, a_k) + K_{k-1}(a_2, \ldots, a_k)\big)K_d(a_1, \ldots, a_d) \\
&\quad - K_k(a_1, \ldots, a_k)\big(a_0 K_d(a_1, \ldots, a_d) + K_{d-1}(a_2, \ldots, a_d)\big) \\
&= K_{k-1}(a_2, \ldots, a_k)K_d(a_1, \ldots, a_d) - K_k(a_1, \ldots, a_k)K_{d-1}(a_2, \ldots, a_d).
\end{aligned}
$$

We then recognize the last line as an instance of the identity for $k$, depth $d - 1$, and sequence $a_1, \ldots, a_d$. Applying the inductive assumption,

$$
\begin{aligned}
&K_{k+1}(a_0, \ldots, a_k)K_d(a_1, \ldots, a_d) - K_k(a_1, \ldots, a_k)K_{d+1}(a_0, \ldots, a_d) \\
&= -(-1)^k K_{d-1-k}(a_{k+2}, \ldots, a_d) \\
&= (-1)^{k+1} K_{d-(k+1)}(a_{(k+1)+1}, \ldots, a_d),
\end{aligned}
$$

as required. $\qquad\square$

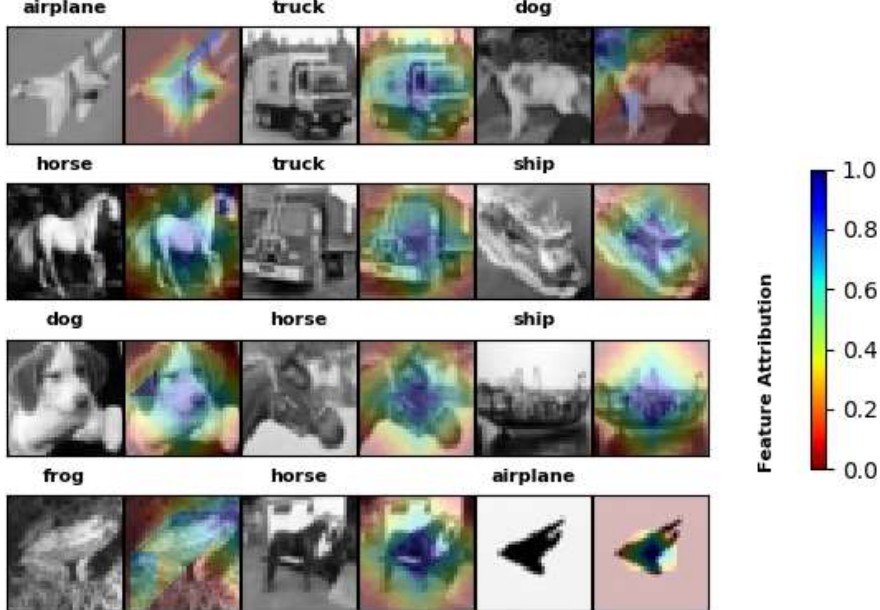

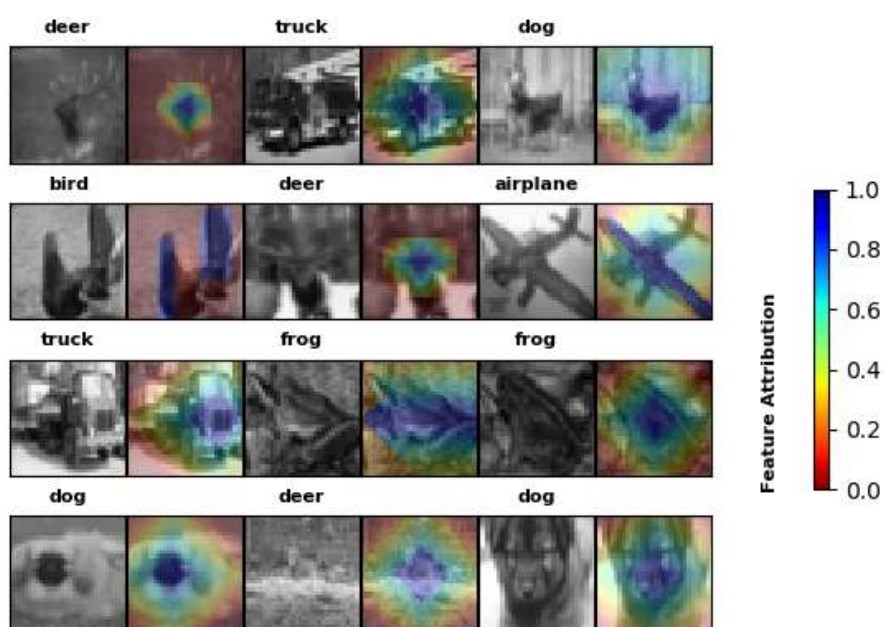

Figure 6: Above we see 24 randomly chosen CIFAR10 test images (in grey scale) and to the immediate right of each their corresponding (normalized) attributions overlayed as a colormap over each of them using the IC strategy. We see that in many cases meaningful aspects are highlighted as important (blue color) in the respective images such as wings for airplanes, face and body parts for animals and frontal frame for trucks.

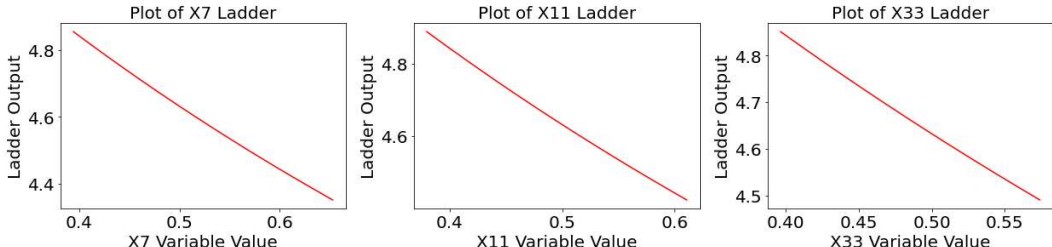

Figure 7: Above we see plots of the functions that represent the three most important variables for the Waveform Dataset: X7, X11, and X33.

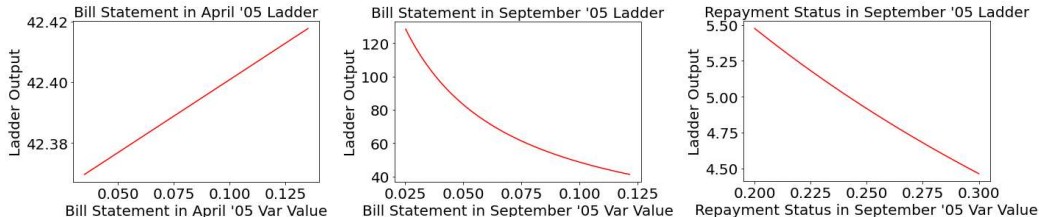

Figure 8: Above we see plots of the functions that represent the three most important variables for the Credit Card Dataset: Amount of Bill Statement in April 2005, Repayment Status in September 2005 and Amount of Bill Statement in September 2005.

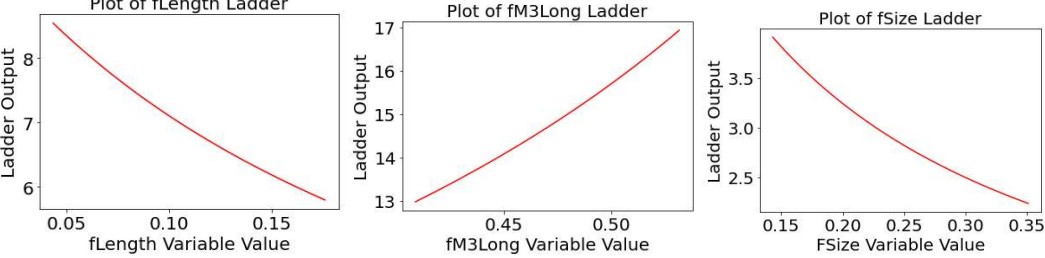

Figure 9: Above we see plots of the functions that represent the three most important variables for the MAGIC Telescope Dataset: FLength, FM3Long and FSize.