# OpenReview forum: "CoFrNets: Interpretable Neural Architecture Inspired by Continued Fractions"
_NeurIPS.cc/2021/Conference — NeurIPS 2021 Poster_

### Official Review · Reviewer_ByEC · 2021-07-16

**Rating:** 5
**Confidence:** 4

**Summary:**

The paper introduces CoFrNet, a new neural net architecture inspired by continued fractions. Several different training strategies are suggested, such as joint training, incrementally fit each layer, etc. Synthetic data and real data experiments are provided, and accuracy comparisons compared to several interpretable generalized additive models, trees, neural nets, etc. are provided

**Limitations And Societal Impact:**

I don't have any concerns here.

**Main Review:**

The ideas presented in this paper are interesting. I would like to clarify some questions about the evaluation:
-	Are the numbers in Table 1 accuracies or AUCs? The caption says accuracies but just wanted to confirm. If some of these datasets are imbalanced, accuracies would not be a good measure.
-	Why does MLP have such a low accuracy on Waveform (0.34), a lot lower than the other methods where the lowest accuracy was 0.69?
-	That the GAM and EBM results are exactly the same for the 3 tabular datasets (Waveform, Magic, Credit Card) seems a little unusual to me because the pygam package uses different types of spline base learners, if I’m not mistaken, while EBM uses a tree base learner, and previous work in the literature (e.g. http://citeseerx.ist.psu.edu/viewdoc/download?doi=10.1.1.433.8241&rep=rep1&type=pdf which first introduced EBMs) showed that the choice of base learner can matter a lot.
-	Please provide standard deviations for Table 1 so that the reader can tell if the results are statistically significant.

Since interpretability is one of the goals here, it would be nice to provide some visualizations of the output that humans can interpret. It appears that CoFrNet-D can be visualized as an additive model (one feature at a time), while CoFrNet-F and CoFrNet-DL cannot. However CoFrNet-D seems to have worse performance than other additive models like EBM, NAM, etc.

Writing:
- “Reference [2] is suited for tabular and image data, 112 whereas [52] is suitable for text data. Reference [19] is a …”  seems a little non-standard to say Reference [x], maybe use author names, e.g. name et al. …..?

**Time Spent Reviewing:**

1

---

> ### Author Response · Authors · 2021-08-09
> **Response to Reviewer  ByEC**
>
> 1) **Accuracies or AUCs:** The reported numbers are accuracies as the datasets are close to being balanced.
>
> 2) **MLP performance on Waveform:** We searched the best MLP model up to depth 25 for all datasets where width was varied from 3 to 15. The architectures performed reasonably in general so not exactly sure why it did worse on Waveform.
>
> 3) **GAM vs EBM performance:** While the results might seem a bit unusual, we did run both methods as indicated in the readme of their open source packages referenced in the paper. For GAMs, we adapted the example they provided in their library (github.com/dswah/pyGAM) and for EBMs, we used the fit function (ebm.fit(Xtrain, ytrain)) that they provided (github.com/interpretml/interpret).
>
> 4) **Statistical significance of results and standard deviations:** The results are statistically significant as we performed paired t-tests (mentioned in the caption) to determine significance, except for CIFAR-10 which came with a fixed test set. This is why two different accuracies have been bolded in some cases in the same column in Table 1 as the difference wasn't statistically significant. Regarding standard deviations, your point is well taken and we will provide them in the final version. The standard deviations ranged between $0.003$ and $0.012$ for all datasets and methods (except CIFAR-10).
>
> 5) **Interpretability of CoFrNet architectures:** Both CoFrNet-D and CoFrNet-DL have additive components that can be visualized. The latter also captures interactions, which could also be visualized up to order 2. We will add such plots in the final version as NeurIPS allows an extra page. Moreover, all three variants, including CoFrNet-F, can be interpreted using the IC and IPS strategies described in lines 173-205 and showcased in section 6.1, whereby one can obtain univariate feature attributions as well as attributions for interactions. This is because all the variants conform to the continued fraction functional form.
>
> 6) We will address your other minor comments

---

> > ### Author Response · Authors · 2021-08-27
> > **More evidence showing interpretability...**
> >
> > You can now find explanations using the IC strategy for 24 randomly chosen CIFAR10 test images [here](https://drive.google.com/file/d/1IbGeUUPUNjl0P_NC-tAPdvIsYiydnfou/view?usp=sharing) for the CoFrNet-DL model. We have also added function plots (as is typically done for additive models such as GAMs) for the top 3 features for each of the tabular datasets and word clouds for the two text datasets showcasing the fact that our model is also globally interpretable.

---

### Official Review · Reviewer_bQJD · 2021-07-16

**Rating:** 8
**Confidence:** 3

**Summary:**

The authors propose three variants of a novel neural architecture based on continued fractions, where the coefficients are learned linear combinations of features.
The authors claim global intepretability of this architecture through the additive nature of one variant and introduce two approaches exploiting the architecture's analytical form to compute first and higher order feature attributions.
They provide an original theoretical proof that a linear combination of continued fractions, i.e. their architecture, is a universal approximator.
Finally, the authors demonstrate the performance and interpretability of the architecture on a 2-dimensional toy experiment, as well as on multiple real-world data sets compared to other models.

**Limitations And Societal Impact:**

Some limitations are discussed with respect to with the architecture's practical applicability.
Societal impact is left unaddressed in the paper.
A very short remark in Section 7 to its (and other globally interpretable neural network architectures) impact may be beneficial.


**Main Review:**

### Originality:
Continued Fractions are well-known, and to the best of my knowledge, this is the first time they were suggested for designing a neural architecture.
As far as I can tell, the interpretability the authors provide by its functional form has not been done like this before.
While I must admit that I had difficulties following the proof for universal approximation, it is original as far as I can tell.

### Quality:
The quality of this paper is quite high, and the work feels well polished.
This work motivates the introduced neural architecture well, and provides not only a wide range of experiments to justify its existence, but also proves its property as a universal approximator.
The authors discuss weaknesses of their architecture.

Since the authors put a strong emphasis on the interpretability aspect of their architecture, I find the demonstration thereof somewhat short-coming in the manuscript.
Given the strong emphasis, I would have liked to see a more quantitative analysis of the interpretability aspect, especially compared to the other globally interpretable models.


### Clarity:
I find the manuscript is well written and well organized.
Beyond technicality, the manuscript is relatively easy to follow.
Methods are introduced adequately.

### Significance:
The architecture is a new approach for globally interpretable models, based on the well-known continued fractions.
Although the manuscript is limited in its practical demonstration of the architecture's iterpretability,
I believe it provides a foundation for equally interesting research and demonstrates a new instance to prove universal approximation.



**Time Spent Reviewing:**

5

---

> ### Author Response · Authors · 2021-08-09
> **Response to Reviewer bQJD**
>
> We are glad you found our work interesting. We will definitely add more information (viz. move feature attributions for Waveform and Quora from supplement to main paper, individual variable function plots, etc.) to bring out the interpretability aspect more strongly. We are not completely certain of what you mean by quantitative analysis of interpretability, but we can say the following things:
>
> 1) Our CoFrNet-D and CoFrNet-DL architectures can minimally be interpreted in the same manner that additive models such as GAMs, NAMs and EBMs can be interpreted by either observing the feature attributions (as reported in the paper) or by looking at the univariate function plots (like the left panel of Figure 2 of arXiv:1909.09223).
>
> 2) All three variants which includes CoFrNet-F can be interpreted using IC and IPS strategies described in lines 173-205 and showcased in section 6.1, where one can obtain univariate feature attributions as well as attributions for interactions since, all the variants conform to the continued fraction functional form.
>
> Hence, the additive architectures in our case can be interpreted in more ways than standard additive models where we exploit their specific functional form. We will also add a remark about social impact, where inherently interpretable architectures are significantly more trustworthy in high stakes decision making (Rudin, Stop explaining black box machine learning models for high stakes decision making, Nature Mach. Intelligence 2019) as opposed to explaining black box models.

---

> > ### Author Response · Authors · 2021-08-27
> > **More evidence of interpretability...**
> >
> > You can now find explanations using the IC strategy for 24 randomly chosen CIFAR10 test images [here](https://drive.google.com/file/d/1IbGeUUPUNjl0P_NC-tAPdvIsYiydnfou/view?usp=sharing) for the CoFrNet-DL model. We have also added function plots (as is typically done for additive models such as GAMs) for the top 3 features for each of the tabular datasets and word clouds for the two text datasets showcasing the fact that our model is also globally interpretable.

---

> > > ### Comment · Reviewer_bQJD · 2021-08-30
> > > **Satisfied, highly positive score unchanged**
> > >
> > > Thank you for the additional work and detailed replies to all the reviewers.
> > > While I feel my initial score may have been very optimistic, I think with the
> > > new additions the score is justified.
> > >
> > > The results on interpretability on Cifar10 seem to point into the right direction.
> > >
> > > I think especially the results on ImageNet show the great potential of CoFrNet
> > > with such a low amount of parameters at about the level of MobileNetv3-small,
> > > but still with better top-1 and top-5 accuracies.

---

> > > > ### Author Response · Authors · 2021-08-30
> > > > **Thank you**
> > > >
> > > > Thank you very much for continuing to see the potential in our idea.

---

### Official Review · Reviewer_Sp1r · 2021-07-21

**Rating:** 8
**Confidence:** 3

**Summary:**

The authors propose a new neural architecture inspired by continued fractions, which has some great theoretical properties in terms of function approximation. As a neural architecture, CoFrNets can learn function with much less number of parameters compared to fully connected neural networks. The authors also claims that CoFrNets is interpretable since it can represented as a polynomial.

**Limitations And Societal Impact:**

yes, but not enough, see main review

**Main Review:**

The idea of considering continued fractions for neural networks is novel, however, the claim that the model is interpretable is not supported well. In (1), the authors discussed first-order attributions and high-order attributions. Arguably, if one wants to get a sparse linear/ polynomial function, learning linear models on polynomial expansions and kernel methods would be what comes to mind. I would argue that these two have at least the same explainability to CoFrNets, but they are not compared to.

Some theoretical discussion may also be added compared to polynomial functions. It is mentioned that the rational approximations formed by any of its finite truncations (termed convergents) are closer to the true value than any other rational number with the same or smaller denominator. However, this is not useful as a function approximator. If the Ground Truth is a high order polynomial of order k with d terms, a polynomial expansion would need to represent the function with (# features)^k input features with d non-zero term. Thus, the number of parameter here is  (# features)^k, and the actual parameter needed is d. In this case, can you prove the number of parameters needed to accurately estimate the polynomial function? It seems like you definitely need less than (# features)^k parameters (although I don't know if polynomials are a subset of continuous fractions). Some analysis will also be helpful to readers unfamiliar with continued fractions.

To raise the quality of the work, I would consider 1. convolution CoFrNets, where each ladder only receives the input dimensions that are structurally similar. 2. using CoFrNets in existing NN layers, so that it is comparable with SOTA models. If CoFrNets is only replacing the most memory-heavy component of some existing model, it might make the model more lightweight.

The model architecture kind of reminds me of transformer models, and by the self-attention mechanism, it would be much easier to construct polynomial functions for transformers. However, I have doubts that whether continuous fractions is better at learning functions that are not polynomial (see sin cos example in appendix). I feel like the authors did not do enough empirical studies to understand why CoFrNets is better or worse compared to MLP. Universal Approximation already exists for MLPs, and NNs are known to be notoriously over-parameterized (which is one of the reasons that it succeeds), and thus outperforming NNs with the same number of parameters is not really impressive. I would like to understand what is the function space where CoFrNets performs better, and what is the function space where CoFrNets performs better. For example, in image classification CNN will perform better than CoFrNet even if CoFrNet has the same number of parameters compared to CNN (at least for now). If for any function CoFrNets outperforms MLP with "any'' number of parameters (not just the parameters cherry picked for CoFrNets), I will be very surprised. I would like to see the performance of CoFrNet and MLP both varying the number of parameters (instead of only one set of parameters which is probably cherry picked for CoFrNets). Also, I would like to see more function classes being compared to. For example, you can set the ground truth to be some random forest regression output, and compare how well MLP and CoFrNet compare since most function spaces (estimated in practice) are not polynomials (at least not with sparse parameters). Discussing and understanding when CoFrNet performs worse than MLP (and also when it is better) is going to make the paper more valuable .

--------------------------------------------------------------------------------

After the authors showed results for imagenet I have faith for this method to become very impactful in the future. This is a very clear accept in my opinion. The interpretability claim is not needed to make this a strong work as the potential of the new model architecture is very high. More theories and applications can be developed along this type of research.



**Time Spent Reviewing:**

5

---

> ### Author Response · Authors · 2021-08-09
> **Response to Reviewer Sp1r**
>
>
> 1) **About sparse polynomial functions:**
> If one wants to represent a sparse polynomial in $p$ variables with at most $k$ terms, each of degree at most $d$, then going by the proof of Theorem 2 (our representation theorem), you only need $2k+1$ ladders, each of depth $d$, and there is a member of the CoFrNet class with no more than these dimensions that can represent the sparse polynomial. This  potentially saves on the representational burden of listing out $p^d$ features (monomials) and then finding a sparse linear combination as suggested by the reviewer. Doing a detailed study of learnability of sparse polynomials leveraging this potential representational benefit would be future work. Please refer to lines 265-270 in the paper for this comment.
>
> 2) **Representing polynomials:** The proof of our representation theorem (Theorem 2) indicates that the class of functions represented by CoFrNets is a superset of polynomials of linear functions (which are also dense in the space of continuous functions like polynomials), which implies we should be able to compactly represent polynomial functions where every linear functional is just a single variable.
>
> 3) **CoFrNet Interpretability:** In addition to the IPS strategy we have also shown that CoFrNets can be interpreted using the IC strategy in Proposition 1. Both CoFrNet-D and CoFrNet-DL have additive components that can be visualized. The latter also captures interactions, which could also be visualized up to order 2. We will add such plots in the final version as NeurIPS allows an extra page.
>
> 4) **CoFrNet extensions:** Thanks for the interesting suggestions about merging our architecture with other well known ones. It is something we have alluded to in the paper (lines 370-375) and will be actively exploring. However, before we try these alternatives, we are currently looking at different training strategies to get the most out of the existing CoFrNet architecture so as to keep its advantages such as interpretability, efficient inference, etc. In this paper since we are for the first time introducing this architecture we wanted to test it in its most pure form, where adding known architectural components might make it more of a challenge to ascertain the precise benefit of our contributions.
>
> 5) **CoFrNet vs MLP comparison:** In the synthetic experiments the number of parameters were not cherry picked. If the function was a polynomial, the depth of the networks was set equal to the degree, else to six (lines 544-545 in supplement). Following that, the MLP width was set so as to have a similar number of parameters as our model. For real data, no such restrictions were placed on MLPs and different configurations were tried to obtain the best results. Your larger point though is well taken. It is indeed interesting to understand the exact settings where our architecture would outperform others consistently, and this is something we will definitely look into.

---

> > ### Comment · Reviewer_Sp1r · 2021-08-18
> > **Thanks for the response**
> >
> > Thanks for the response. I remain my positive attitude towards this paper, but I could only raise my score if more analysis is done. I really like the suggestion by reviewer hnfL, to demonstrate a decent accuracy on Imagenet (such as the accuracy for Alexnet) would make a strong case for the potential for CoFrNet. In that case I will raise my score to 8.
> >
> > From figure 3 I do not understand why the IC strategy yields good result. In general, I am much more excited about CoFrNet's potential for efficient representation than interpretability. I really hope the authors better understand when does CoFrNet outperforms MLP in general. In such cases, users would know when to choose CoFrNet over MLP, which will make a strong case to CoFrNet's significance. For instance, we know that CNN outperforms MLP by better capturing the locality of the structure data, and thus CNN should be chosen over MLP when the data has strong locality structure. After looking at the results, I do not understand why and when does CoFrNet outperforms MLP. If you vary parameters with CoFrNet and MLP, does the best CoFrNet still outperform MLP? I want to understand does CoFrNet capture some special structure that is difficult for MLP, or is it just more efficient in parameters (and such advantage does not exist when more parameters are used by MLP).

---

> > > ### Author Response · Authors · 2021-08-27
> > > **We have now experimented on ImageNet...**
> > >
> > > 1. We have now run our model CoFrNet-DL on Imagenet where we were able to get a Top-1 accuracy = 69.7% and Top-5 accuracy = 88.6% (close to Resnet-18 which has top-1 accuracy of 70.09% as per here https://paperswithcode.com/sota/image-classification-on-imagenet). This is better than some other popular architectures such as Alexnet which has Top-1 accuracy = 63.3% and Top-5 accuracy = 84.6%.
> > >
> > > 2. In Figure 3, the IC strategy gives the same first order attributions as the IPS strategy where in both cases they are close to 0 which is consistent with the original functions that have insignificant first order attributions. Since, IC can provide only first order attributions we believe this to be a positive result. In addition, we now have created IC explanations for CIFAR10 [here](https://drive.google.com/file/d/1IbGeUUPUNjl0P_NC-tAPdvIsYiydnfou/view?usp=sharing) which seem to be quite reasonable, along with showcasing global interpretability through function plots for important features and word clouds for tabular and text data respectively.
> > >
> > > 3. For real data, we did not try to match depth or parameters of MLPs with our model, but rather tried different configurations to obtain the best results. Your larger point though is well taken. It is indeed interesting to understand the exact settings where our architecture would outperform others consistently, and this is something we will definitely explore more.

---

> > > > ### Comment · Reviewer_Sp1r · 2021-08-27
> > > > **This is wonderful, some details**
> > > >
> > > > Can you state the number of parameters and more details to train the Imagenet with CoFrNet-DL?
> > > > Is it possible to share the model architecture code? I just want to understand how much existing modules are used and how much more new ones are added. What are the training strategy that makes this work? Either way I view this as a very positive result and will raise my score.

---

> > > > > ### Author Response · Authors · 2021-08-27
> > > > > **More details...**
> > > > >
> > > > > We used as many univariate ladders as the dimensionality of ImageNet images (224x224) along with up to depth 12 of the full ladders which led to ~2.7M parameters. We didn't use any other architectural components other than our architecture and code for which was submitted as part of the supplement. We used early stopping and Adam optimizer where learning rate was set to $e^{-3}$ and weight decay was set to $0.001$. The batch size was set to 128. The number of epochs was set to a maximum of 100 and training was done with dropout. Looking ahead we believe maxout and boosting type ideas along with layer dropout might be promising to try, before we also start augmenting the architecture with known components such as convolutional layers or attention etc. along the lines you suggested.

---

> > > > > > ### Comment · Reviewer_Sp1r · 2021-08-27
> > > > > > **Great result, clear accept**
> > > > > >
> > > > > > I'm raising my score to 8 and I think this paper has great potential in the future. One question is that why is the Cifar result so bad compared to imagenet, as it should be much easier to reach a better performance on Cifar.

---

> > > > > > > ### Author Response · Authors · 2021-08-28
> > > > > > > **Thank you**
> > > > > > >
> > > > > > > Thank you so much for appreciating our work. Regarding CIFAR10 we will try to analyze more. Although it is still much better than other interpretable models.

---

> > > > > > > > ### Comment · Reviewer_Sp1r · 2021-08-30
> > > > > > > > **Clarify results of Cifar-10**
> > > > > > > >
> > > > > > > > I want to strongly urge the authors to redo experiments on Cifar. The result for cifar-10 and imagenet having the same top-1 accuracy is highly unreasonable (unless it's due to CoFrNets do not work well on small images). I trust the authors on their result in imagenet, but for general readers this weird result may be a red flag for the correctness of the paper. If I saw a paper that says they get 73 accuracy on Cifar-10 and 70 top-1 accuracy on imagenet, I would be very doubtful of the result. My best guess is the model for cifar-10 is under-parameterized, or some data augmentation methods is needed for training. To redo the Cifar-10 result, the authors do not need to compare with other interpretable methods and do not need to limit the model size. I like to see the result for Cifar-10 using similar strategies to make imagenet work.
> > > > > > > >
> > > > > > > > Either releasing the code, or redo experiments with more parameters for Cifar-10, or (understanding and) stating that CoFrNets is unsuitable for small-scale image classification are all methods to address the potential doubts. However, I urge the authors to do this before the end of rebuttal period (or at the very least before when the AC decides to accept the paper). In the conference review period reviewers are not responsible (and cannot) check the correctness of the experiment. However, I hope the authors can address this major issue without reviewers threatening to lower the scores.

---

> > > > > > > > > ### Author Response · Authors · 2021-08-30
> > > > > > > > > **Yes, we will rerun on CIFAR10 with Imagenet setup**
> > > > > > > > >
> > > > > > > > > We were anyway thinking of doing what you suggested for the final submission i.e. rerun on CIFAR10 using Imagenet setup and if nothing changes then try to see why that is happening. We will report results once we have them, hopefully in the next few days.

---

> > > > > > > > > > ### Author Response · Authors · 2021-09-02
> > > > > > > > > > **Reran experiments on CIFAR10**
> > > > > > > > > >
> > > > > > > > > > We ran CoFrNet-DL with the parameter setup described for ImageNet above under two settings: 1) Train directly on CIFAR10 and 2) Train with data augmentation (like you suggested) where the specific data augmentation we apply is of randomly cropping and flipping images. With the former we obtained an accuracy of 76.4% and with the latter we obtained an accuracy of 87.2%. Hence, the new setup with data augmentation seems to help. We will update the results in the paper based on this new finding.

---

### Official Review · Reviewer_hnfL · 2021-07-22

**Rating:** 6
**Confidence:** 3

**Summary:**

The paper proposes CoFrNet, a neural architecture that stacks neural layers in a way inspired by Continuous Fractions. The paper shows some interpretability advantage of the proposed model.

**Limitations And Societal Impact:**

Limitations are discussed adequately. I don't find any discussion on societal impact but I guess there is no need.



**Main Review:**

Pros:
1. There is some novelty as this is (to my best knowledge) the first work that combines Continuous Fractions with Neural Networks.
2. Theoretical development on the interpretability via Power Series and universal approximation seems sound.
3. The paper is in general clearly written.

Cons:
1. While the paper does show the interpretability advantages on synthetic functions and some of the datasets tested, discussion on other datasets ( such as Cifar-10) is missing.
2. I also feel that the interpretability discussion on non-synthetic datasets is very limited. The authors only listed a few top features. I think a more thorough analysis, possibly including plots illustrating the feature attributions, is needed.
3. The performance results (in term of accuracy) are poor compared to SOTA models. While the proposed CoFRNet does have interpretability advantages, the poor performance will not allow it to be used in any real-world scenarios. I also doubt if CoFRNet can be applied to more difficult datasets (such as ImageNet) as the inverse activation will have serious gradient variance issues when you stack many more layers to build a deeper model.




**Time Spent Reviewing:**

3

---

> ### Author Response · Authors · 2021-08-09
> **Response to Reviewer hnfL**
>
> 1) **Showcasing Interpretability:** First, important features for Waveform and Quora are already mentioned in the supplement (lines 537-541). Second, we can provide univariate function plots for individual variables (like the left panel of Figure 2 of arXiv:1909.09223) in the final version (NeurIPS allows an extra page). These go beyond feature attribution plots in showing the shape of the dependence on each feature, in addition to magnitude and sign. They are readily available for the CoFrNet-DL model, which is additive with some interaction terms and was best performing. As for CIFAR-10, since we are talking about global interpretability, we did not show heatmaps but we could add example-specific explanations using the IC strategy.
>
> 2) **CoFrNet Performance:** Although we acknowledge that CoFrNet results are some ways away from SOTA on image and text datasets currently, it also is equally evident that we are considerably better than SOTA interpretable models (including the recent LassoNet, see common response above), which we believe is a significant contribution in itself. As mentioned in the paper we are looking at different training strategies (layer dropout, maxout, boosting, etc.) currently to narrow this gap. However, besides the favorable comparison to other interpretable models, the architecture also has other attractive properties such as the possibility of extremely efficient inference where the coefficient vectors in each layer can be batched and multiplied with the same input that is passed to every layer. This is not possible for standard NN architectures as the representation fed into each layer changes.
>
> 3) **Exploding gradients for training deep CoFrNets:**
>
> a) First, it is important to realize that our function class is very different than that represented by standard NN architectures. The representation theorem (Theorem 2) in our paper indicates that having more ladders is more important in covering a larger span of functions than building deeper ones. Increasing the depth beyond a point only results in some fine tuning which may have minimal overall effect. Intuitively, the reason for this to happen is that each ladder when flattened/collapsed represents a rational function of degree equal to the depth $d$, but linearly combining $L$ of these -- which is what happens when we train an ensemble of these -- leads to a degree $Ld$ rational function when reduced to a single one (assuming that the ladders are not exactly the same). Hence, rich functions can be represented even without having to build deep ladders.
>
> b) Second, our activation function given on line 169 in the paper clips the $\frac{1}{x}$ function and hence by extension can also clip the gradient sufficiently for appropriate choices of $\epsilon$. Gradient clipping is a standard procedure to address the exploding gradients issue in neural networks (Zhang et, al. Why gradient clipping accelerates training: A theoretical justification for adaptivity, ICLR 2020).
>
> c) Third, an interesting direction we are exploring is that our interpretation strategy based on continuants in Proposition 1 (line 181) can be used to compute gradients without backpropagation/chain rule. Hence it could potentially be used to also train our model, using a similar expression that can be derived for gradients w.r.t. $w$. This would also address the exploding gradient problem.

---

> > ### Comment · Reviewer_hnfL · 2021-08-17
> > **Reply**
> >
> > Thank you for your reply. While some of my doubts are solved, I am still not convinced about these two points:
> > 1. The authors said example-specific explanations using the IC strategy can be added for Cifar-10. Can authors perform this analysis and show some results?
> > 2. In terms of training deep CoFrNet, while I agree that the laddered structure can represent a rich set of functions, I am not convinced that it can represent functions that need to be hierarchical and deep, such as the object recognition functions needed in Computer Vision tasks. I think this is also probably why CoFrNet has such poor performances on Cifar-10. I think the best way to prove me wrong is to perform experiments on some larger datasets (e.g. ImageNet) and show some decent performances (Doesn't have to be SOTA).
> >
> > I would be happy to raise my score if the authors can do the above.

---

> > > ### Author Response · Authors · 2021-08-27
> > > **Answering both your questions...**
> > >
> > > 1. You can now find explanations using the IC strategy for 24 randomly chosen CIFAR10 test images [here](https://drive.google.com/file/d/1IbGeUUPUNjl0P_NC-tAPdvIsYiydnfou/view?usp=sharing). We have also added function plots (as is typically done for additive models such as GAMs) for the top 3 features for each of the tabular datasets and word clouds for the two text datasets showcasing the fact that our model is also globally interpretable.
> > >
> > > 2. For your second ask we have now run our model CoFrNet-DL on Imagenet where we were able to get a Top-1 accuracy = 69.7% and Top-5 accuracy = 88.6% (close to Resnet-18 which has top-1 accuracy of 70.09% as per here https://paperswithcode.com/sota/image-classification-on-imagenet). This is better than some other popular architectures such as Alexnet which has Top-1 accuracy = 63.3% and Top-5 accuracy = 84.6%. Although some ways away from SOTA, we believe this to be decent performance and given that our architecture is simple, compact, efficient for inference, and interpretable, we hope you will consider it to be of value, not to mention that a lot more improvement may be possible as we try more training strategies mentioned before.

---

> > > > ### Comment · Reviewer_hnfL · 2021-08-30
> > > > **Raising Score**
> > > >
> > > > Thank you for your efforts in the extra experiments. This indeed gives more confidence, even though that the ImageNet performance is still far from that of modern architectures. As promised I will raise my score.

---

### Author Response · Authors · 2021-08-09
**Common Response**

We would like to thank all the reviewers for their constructive comments. We are glad that all of you found our work to be novel and interesting.

Post submission, we were also able to test another very recent approach that claims to be an interpretable neural architecture termed as LassoNet (Lemhadri, Ruan, Abraham and Tibshirani, LassoNet: A Neural Network with Feature Sparsity, AISTATS 2021). We now report its performance along with rest of the entries (except black box SOTA) in Table 1 for convenience. We will update Table 1 based on this in the final version of the paper. As can be seen CoFrNet significantly outperforms this model too on most datasets.

| Methods | Interpretable | Waveform | Magic | Credit Card | CIFAR10 | Sentiment | Quora |
| ---------- | ------------- | -------- | ----- | ---------- | ------- | --------- | ----- |
| CoFrNet-DL | Yes | 0.81 | **0.84** | 0.69 | **0.73** | **0.84** | **0.88** |
| CoFrNet-D | Yes | 0.69 | 0.76 | 0.66 | 0.38 | 0.80 | 0.75 |
| GAM | Yes | **0.85** | **0.85** | **0.72** | DNC | 0.51 | DNC |
| NAM | Yes | **0.86** | 0.81 | 0.69 | 0.38 | 0.50 | 0.49 |
| EBM | Yes | **0.85** | **0.85** | **0.72** | 0.40 | 0.59 | 0.49 |
| CART | Yes | 0.75 | 0.79 | 0.69 | 0.29 | 0.52 | 0.73 |
| *LassoNet* | Yes | 0.84 | 0.76 | 0.67 | 0.28 | 0.50 | 0.53 |
| MLP | No | 0.34 | 0.65 | 0.50 | 0.35 | **0.83** | 0.85 |

We now address specific comments of the reviewers.

---

### Decision · Program_Chairs · 2021-09-27

**Decision:**

Accept (Poster)

**Comment:**

The reviewers have agreed on many positive aspects of the paper and the authors have done a great job in addressing the concerns. The paper is on a highly important topic and contains novel ideas.

There were many important points in the reviewer responses that should be integrated into the final version of the paper.